# Environmental Impacts and Strategies for Bioremediation of Dye-Containing Wastewater

**DOI:** 10.3390/bioengineering12101043

**Published:** 2025-09-28

**Authors:** Mukesh Kumar, Anshuman Mishra, Suresh Kumar Patel, Jyoti Kushwaha, Sunita Singh, Vinay Mishra, Deepak Singh, Vijay Singh, Balendu Shekher Giri, Reeta Rani Singhania, Dhananjay Singh

**Affiliations:** 1Department of Chemical Engineering, Institute of Engineering & Technology Lucknow, Luvknow UP-226021, India; mukesh_ivrcl@rediffmail.com (M.K.); jyotikushwaha111213@gmail.com (J.K.); vinay.mishraa@gmail.com (V.M.); dsdeepaketawah8@gmail.com (D.S.); 2Department of Chemical Engineering, BTKIT, Dwarahat UK-263653, India; anshubiet1@gmail.com; 3Department of Chemical Engineering, GPL, Lakhimpur-Kheri UP-262701, India; suresh.ieju@gmail.com; 4Department of Pharmacy, Ambekeshwar Institute of Pharmaceutical Sciences Lucknow, Lucknow UP-226201, India; sunita.pharma80@gmail.com; 5Centre for Advanced Studies, Dr. APJ Abdul Kalam Technical University, Lucknow UP-226031, India; vijch27@gmail.com; 6Sustainability Cluster, University of Petroleum & Energy Studies, Dehradun UK-248007, India; 7Institute of Aquatic Science and Technology, National Kaohsiung University of Science and Technology, Kaohsiung City 81157, Taiwan; reetasinghania@nkust.edu.tw; 8Rajkiya Engineering College, Ambedkar Nagar UP-224122, India

**Keywords:** bioremediation, bioreactor, dye, microbial fuel cell, textile wastewater

## Abstract

Rapid industrialization, along with the development of textile and other associated industries, has led to the discharge of dyes, heavy metals, and other carcinogenic and environmentally harmful substances into water bodies. The volume of wastewater containing dyes is increasing day by day. Raised levels of dyes, along with other contaminants, in wastewater are becoming a global concern, as these affect human health as well as aquatic flora and fauna. Bioremediation is one of the effective, sustainable, eco-friendly and cost-effective approaches for the treatment of wastewater containing dyes. This paper presents a state-of-the-art review of bioremediation techniques used for the removal of dyes from textile wastewater. The usage of various strains, e.g., bacteria, algae, yeast, enzymes, fungi, etc., is discussed in detail. Bioremediation of dyes using bioreactors and microbial fuel cells is also explored in this study.

## 1. Introduction

Heavy volumes of intensively colored wastewater from textile and dye manufacturing industries constitute a major environmental barrier. Textile manufacturing industries require high amounts (thousands of gallons) of water every day. The effluent is polluted with contaminants like detergents, suspended solids, heavy metals, insoluble substances, dyes, additives, aldehydes, non-biodegradable matter, etc. [1]. More than ten thousand types of dyes and pigments are consumed by these industries and the global dye production is estimated to be 7 × 10^5^ tons per year. About 10% of dyes used in industries are discharged into wastewater [2,3].

Pollution due to dye discharge from textile industries poses a potential threat to ecosystem. Dyes are regarded as hazardous contaminants by agencies like the United States Environmental Protection Agency (USEPA) and the Occupational Safety and Health Administration (OSHA). Textile industry pollutants usually comprise dyes along with large amounts of biochemical oxygen demand (BOD), chemical oxygen demand (COD), total dissolved solids (TDS), total suspended solids (TSS) and heavy metal traces [4]. Due to the carcinogenic nature of dye, its presence in small amounts can also lead to a huge environmental impact. This increases the requirement for the development of new treatment techniques which are cost-effective, simple to use, efficient and environmentally friendly [5,6]. Various biochemical and physicochemical processes like adsorption, precipitation, flocculation, ion exchange, membrane separation, filtration, coagulation, oxidation, and microbial degradation are generally used for treatment of wastewater containing dyes. These traditional techniques suffer from severe limitations, e.g., lesser efficiency, higher operation cost, complex steps, high quantity of sludge formation and less scalability to a commercial level [7,8,9].

Bioremediation has emerged as a promising and ecotoxicological technique for the treatment of dye-based wastewater [10]. This technique uses microorganisms to degrade the chemical structure of textile dyes, which, in turn, leads to transformation, partial degradation and mineralization. Recent studies were more focused on sustainable and biological techniques to lessen the negative effects of textile dye pollution on the environment for wastewater treatment [11,12]. This review focuses on decontamination of dye-based wastewater using various techniques and discusses the techniques involving the usage of microorganisms.

### 1.1. Classification of Dyes

Dyes are basically organic compounds which have capability to attach to fabric in order to impart color on it. Chromophores and auxochromes are the two main compositions of dye molecules. Chromophores are liable for producing colors, while auxochromes act as supplements to chromophores, providing the ability to dissolve in water and enhancing the affinity towards fibers [13]. Dyes are further classified into several different categories on the basis of their color, method of application, origin, solubility and structure. On the basis of origin, dyes can be classified into synthetic and organic types. On the other hand, azin, anthraquinone, chromophoric, nitroso and acridine dyes are the major types on the basis of chemical structure. On the basis of application, dyes can be azoic colors, dispersive dyes and vat dyes. Dyes can also be listed on the basis of their solubility, such as soluble and insoluble. Soluble dyes are further classified depending on their particle charge into anionic and cationic type. Anionic forms consist of acid, reactive, direct, and mordant, while the cationic form consists of basic. Insoluble dyes consist of azo, sulfur, disperse, solvent and vat [12,14]. The classification of dye on the basis of structure, origin and application is shown in Figure 1, and on the basis of solubility, in Figure 2.

Azo dyes consisting of an -N=N- group are frequently used in industries manufacturing textiles and responsible for more than 50% of the global dye production. These dyes can be named as mono-, di-, tri- or poly-azo-dyes on the basis of the azo groups available. Anthraquinone has a very good fastness property. Acetate dyes, also called disperse dyes, are used for dying of nylon, hydrophobic fibers and cellulose acetate.

Dye degradation can occur either aerobically or anaerobically, involving different enzymes, intermediates, and stages—particularly synthetic dyes like azo dyes. Aerobic dye degradation is the process by which bacteria use oxygen to break down dye molecules into less toxic forms such as ammonia, carbon dioxide, and water. Anaerobic dye degradation occurs when enzymes break down the dye’s azo link, producing colorless, possibly hazardous aromatic amines as by-products. It is common practice to employ a sequential anaerobic–aerobic procedure in which the dye is decolored in the anaerobic step and the dangerous amine intermediates are mineralized in the aerobic step [12].

Azo dyes are classified as either harmful or non-toxic based on laboratory tests. Figure 3 illustrates how hazardous dyes cause free and N-acetylated amino groups to emerge in dyes that form RNA/DNA nitrenium bonds, rendering the dye genotoxic. These dyes are primarily carcinogenic, mutagenic, lipophilic, and hydrophobic (absorbed by the bacterial cell and decreased in the cell, causing cellular and ecological consequences). Alkyl and arylamines, as well as the addition of carboxyl and sulfo groups, make non-toxic azo dyes non-carcinogenic.

The ecotoxicity of azo dye is significantly influenced by a number of environmental elements, including biotic and abiotic factors, as well as chemical qualities under natural conditions. The ecotoxicological effects of azo dyes in natural settings, however, have not yet been investigated or hypothesized. Currently, because of the wide and varied structure of azo dyes, it is hard to evaluate each dye’s ecotoxicity without increasing the cost [14].

### 1.2. Harmful Effects of Textile Wastewater

Dye-containing textile wastewater has adverse effects on human life and water bodies. It disturbs the water quality in water bodies, disrupts photosynthesis, affects food chain, and deteriorates the growth of plants, recalcitrance and bioaccumulation, leading to an increase in BOD, COD, toxicity, mutagen city and carcinogenicity [15]. Even small traces of dyes can have an adverse environmental impact because of their carcinogenic nature. The presence of dyes in water reduces the penetration of sunlight into water, leads to color change, and hinders the photosynthesis process, which impacts aquatic flora and fauna [16,17]. The presence of metals and chlorine in the wastewater from textile industries also affects marine life. Due to eutrophication, dye and pigments impact the quality of water and adversely affect the ecological status of aquatic life. Dyes can also impact human health if they come into contact with human organs through the food chain [14,18,19].

### 1.3. Treatment of Dyes

The treatment of dyes, especially from industrial wastewater (like textile dyeing), is a critical environmental concern. Dyes are often toxic, persistent, and resistant to conventional wastewater treatments. There are several methods to treat dye-containing water, and they can be grouped into physical, chemical, and biological processes [20]. These conventional techniques have several disadvantages, like high cost, sludge/membrane disposal, toxic by-products, high operational costs, energy demand, slow, sensitivity to environment, and being ineffective for synthetic/complex dyes. Therefore, bioremediation has gained interest by researchers as it is a cost-effective and better option for the treatment of textile wastewater [21].

#### 1.3.1. Physicochemical Methods

Different physicochemical techniques employed for the treatment of textile wastewater include oxidation, flocculation, adsorption, bleaching, coagulation, ion exchange, membrane filtration and precipitation. The conventional techniques are costly and need large amounts of energy. Their limitations include a huge amount of sludge formation, less feasibility at a commercial scale, and large amounts of chemicals [12,22,23].

##### Adsorption

With the increase in demand of the treatment of wide varieties of effluents, there has been rapid research and development of more effective adsorbents. These are widely used because of their low cost, ease of application and high efficiency. Adsorption has higher decolorization efficiency for different dyes [24]. Different types of sorbents, like activated carbon (AC), chips of wood, carbon nanotubes (CNT), coir pith, fly ash, peat, metal oxide, zeolites, silica gels, rice husk, Mxenes and other materials, were developed for the removal of dye. These led to enhanced performance because of bigger specific surface areas [19]. Research focusing on metal–organic frameworks (MOFs) is underway due to their changeable chemical framework. However, several disadvantages are also observed, e.g., higher cost, larger retention period, side reactions and a lack of potential towards specific types of dyes. As this technique does not involve dye molecule breakdown, this option was found to be less attractive [12,22].

##### Remediation by Oxidation of Dyes

It is one of the most common techniques used for the removal of dyes. It is popular because of its simplicity. Oxidative cleavage present in the aromatic ring of dye molecules is the main driving force for all oxidation mechanisms. Hydrogen peroxide is the main oxidizing agent used in these processes. UV light is used for activation of H_2_O_2_. Different techniques of oxidation, such as ozonation, sodium hypochlorite, Fenton’s reagent, photochemical and electrochemical destruction, have been used. The stability of ozone is dependent on pH, temperature and the salts present [12]. All of these techniques suffer from one or more disadvantages, e.g.,large amounts of sludge formation, toxic by-product formation, and expensive energy and chemicals [25,26].

##### Ion Exchange

In this technique, ion exchange resin passes all the dye molecules until saturation of all the resin exchange sites. But its expensive nature and ineffectiveness in treatment make it less suitable for dispersive dyes. This technique is not widely used because it is useful for a particular number of dyes only. This method is applicable for both kinds of dyes, i.e., cationic and anionic, and offers several advantages, including more efficient eradication of dyes, no regenerative loss of adsorbent, and solvent recovery after usage [24,27].

##### Membrane Separation

Membrane separation techniques like RO, NF and UF are used for the treatment of textile wastewater. These have several advantages, like a lower footprint, lower maintenance requirements, and ease of installation and operation [12]. Because of their high pressure, dye molecules permeate through membranes [22]. The separation mechanism is based on the pressure difference between the permeate and the feed of the membrane. Several efficient membranes have been used for selective permeation of various dye molecules. Membrane techniques also suffer from disadvantages like incapability to treat large amounts of dyes, large capital investment, formation of concentrated residue, and membrane clogging. The instability of this technique in high-pressure conditions also requires further research.

##### Coagulation

Coagulation is an efficient and effective technique for eradication of dyes. This process makes use of ferrous sulphate and ferric chloride, which is more suitable for removing direct dyes. However, this technique is not suitable for acidic dyes. Also, the coagulants used are expensive in nature, adding to the operating cost of the technique, leading to limited applicability. There are certain demerits associated with this technique, such as huge amounts of sludge formation and determination of the coagulant optimum dosage [24,27,28,29].

#### 1.3.2. Biological Methods

Bioremediation serves as an alternative option to the physicochemical methods for textile wastewater treatment due to its advantages like lower operating costs, environmental benignity, and non-toxic and innocuous by-product formation [30]. However, the treatment process is complicated due to use of microorganisms and intractable nature of dyes. Recombinant and naturally occurring microorganisms are used in bioremediation to break down dangerous substances like dyes into harmless substances like carbon dioxide and water under various aerobic and anaerobic conditions [31]. This method may be applied in both ex situ and in situ modes. Various bioremediation techniques used for textile wastewater treatment are discussed in the section below [32].

## 2. Bioremediation of Dyes

By using a bioremediation approach, organic wastes can be reduced to an acceptable concentration or biologically broken down into harmless molecules. The bioremediation process uses microorganisms, whereby they break down environmental pollutants by eating them. Enough nutrients and other essential compounds must be supplied for dangerous materials to break down properly [33,34]. Enzymes aid in the processes involved in metabolism. Different enzymes have demonstrated improved dye degradation capabilities. The state of the environment is crucial for bioremediation as well since it encourages the microbial development that is needed for the procedure [32]. Microbial growth can be increased by varying the environmental conditions, which, in turn, increase the efficiency of degradation for effective bioremediation [35].

### 2.1. Advantages and Limitations of Bioremediation

The bioremediation process has a number of advantages over traditional methods. It is less labor-intensive, more eco-friendly and cost-effective, leads to complete degradation, reduces sludge generation, and involves adaptable microorganisms. It can be completed on location without disturbing other ongoing activities. It also entails completely decontaminating dangerous substances, such as colors, into safe products without the need for dangerous chemicals. Additionally, this method is easy to use, sustainable, energy-saving, and environmentally benign. In this technique, eliminated contaminants are not transferred to other environmental media. This method has certain drawbacks as well, such as being sensitive to environmental conditions and not effective for all dyes/pollutant, its toxicity being able inhibit microbes, requiring monitoring, and being a slow process [12]. Advantages and disadvantages of bioremediation techniques are shown in Figure 4. Various kinds of bioremediation techniques are shown in Figure 5. The development of sophisticated bioremediation methods appropriate for treating wastewater containing complex contaminants needs more research.

### 2.2. Bioremediation of Dye Using Bacterial Strains

The initial stationary phase of growth of microbial community creates enzymes like azoreductase and laccase, which aid in the cleavage of azo linkage, leading to azo dye decolorization. Bacteria, both in a single culture or consortia, have shown better results in dye decolorization. Bacterial consortium has showed better results as compared to pure isolates [36,37].

Khan and Malik [19] used *Pseudomonas entomophilaBS1* as an isolated strain to breakdown reactive black 5 (500 mg/L), achieving 93% efficiency after 120 h of incubation. Anaerobic conditions typically aid in the azo dye’s removal by the breaking of azo links to generate aromatic amines. Nevertheless, more degradation is required to lessen the harmful effects of these aromatic amines because they are mutagenic and carcinogenic in nature. An aerobic treatment system consisting of bacteria is used for further degradation into harmless compounds of these aromatic amines. The pH, structure, contact time, temperature, and dye concentration of the bacteria are the major parameters that affect this process. Most studies were conducted in the temperature range of 30–40 °C, with a pH range of 3–10 and a dye concentration of 20–5000 mg/L.

Lade et al. [38] employed a consortium of pseudomonas species SUK1 and pseudomonas rettgeri strain HSL1 to degrade four different azo dye variants: direct red 81, reactive orange 16, reactive black 5, and disperse red 78. It was observed that the mineralization of dyes produces intermediates such as aromatic amines, which are subsequently broken down by the enzymes *oxygenase* and *hydroxylase* that are generated by bacteria [34]. Xie et al. [39] used the bacterial flora DDMY1 to study the impact of various parameters on the decolorization of reactive black 5 (RB5) in microaerophilic settings. Decolorization achieved using bacterial flora was greater than 70 percent in both acidic and basic environments in a 4.0–9.0 pH range. The highest discoloration achieved in an acidic medium (at 5.0 pH) was 87.45 ± 1.09%, while that in an alkaline medium (at 9.0 pH)was in the range of 88.89 ± 2.56%. With the rapid increment in temperature from 30 to 40 °C, decolorization was also increased. This could be due to the rapid increase in enzyme activity with an increase in temperature up to a specific range. Due to the further increase in temperature, there is an effect on the enzymatic activities and growth of flora DDMY1. It was also noted that the decolorization rate decreases with a NaCl concentration increase, and a similar pattern was observed when the dye concentration was increased.

### 2.3. Bioremediation of Dye Using Fungal Strains

Degrading and mineralizing textile colors has been discovered to be a successful utilization of robust morphological fungi, potent enzymatic machinery, and metabolism that adapts to changing environmental circumstances [24,40]. Metabolic activities are influenced by intra- and extracellular enzymes which can degrade different textile water dyes [12]. *Penicilliumoxalicum* strain SAR-3 was found to degrade direct red 75, direct blue 15 and acid red 183 dyes [41]. Hyphae of fungi comprising extracellular enzymes eradicate dye by surface adsorption, resulting in chemical bond breakage of dye molecules. Azo dye was completely detoxified and mineralized by *Candida* and *Magnusiomyces* fungal strains [35].

White-rot fungi are most extensively used for dye removal due to the presence of non-specific lignin-modifying enzyme. But these suffer from several limitations, e.g., unreliable production of enzymes, requirement of a large-size reactor for complete degradation, lengthier growth phase, and demand of restrictive nitrogen surroundings [42]. It is not recommended to degrade the dye only in presence of a fungal strain for a longer duration (20–30 days) because the system/substrate does not remain stable due to bacteria growth, which dominate the fungi, hampering the degradation of target dyes [43].

Factors affecting dye degradation include the initial growth conditions, temperature, oxygen, pre-treatment, nutrient concentration, pH, wastewater characteristics, and availability of carbon (C) as well as nitrogen (N) compounds [12]. The effect of carbon and nitrogen in the degradation of Congo red (CR), RB5 and acid orange 7 (AO7) dyes using *Curvularia clavate* NZ2 was studied by Neoh et al. [44]. The majority of dyes underwent degradation in the presence or absence of carbon or nitrogen; however, fungi did not break down AO7 dye when a nitrogen source was present. Akar et al. [45] showed acid red 57 (AR57) biosorption upon *Neurosporacrassa* and various parameter effects on the eradication efficiency of AR57. Arunprasath et al. [46] showed the effect of pH, dye concentration, and temperature on the degradation of Malachite green (MG) using the *Lasiodiplodia.* fungal strain. It was able to remove dye within an acidic pH range, while removal efficiency reduced above pH 7. Similarly, sufficient degradation occurred up to 30 °C and reduced thereafter. Degradation efficiency was observed to be inversely proportional to dye concentration.

### 2.4. Bioremediation of Dye Using Algae Strains

Algae are abundant in freshwater and saltwater, and they have the potential to be a sustainable biosorbent. Algae have drawn attention for their ability to degrade pigments found in textile wastewater. Because of their higher surface area and strong binding affinity, they exhibit characteristics like high biosorption and bio-coagulant capacity. According to Ahmad et al. [24], there are three different kinds of biodegradation processes: anoxic (a mix of aerobic and anaerobic), aerobic, and anaerobic. In routine procedures, the anoxic technique has been applied extensively. This technique involves two anaerobic processes: one with high oxygen demand and another with low oxygen demand to treat textile effluents and resulting effluents. Ruben et al., 2024 [47], conducted a study to evaluate how well Chlorella vulgaris removes contaminants from two different actual textile wastewaters (without dilution) and to assess the composition of microalgal biomass for additional valorization (using a circular economy approach). Microalgae flourished well, with average productivities ranging from 78 ± 3 to 112.39 ± 0.07 mg DW L^−1^ d^−1^ and growth rates between 0.234 ± 0.005 and 0.290 ± 0.003 d^−1^. The degradation of textile dyes can be further categorized on the basis of their mechanism of degradation: biosorption and biodegradation. One potential biosorbent for biosorption is microbial biomass, which is a by-product of industrial fermentations. Many groups (amino, phosphate, thiol and carboxyl) available in the wall of microorganisms’ cells bind the molecules of dyes. This binding sequence is quite fast and is completed within several hours [28,48].

Biodegradation is the biological breakdown of organic substances. Complete biodegradation is referred to as mineralization. When organic substances fully decompose, they produce CO_2_ and water. While fungal biodegradation is unrestricted, bacterial biodegradation of textile dyes has the potential to limit substrate fusion. Algae possess good cell wall properties which help in biosorption, electrostatic attractions and complexation. Algal cell surface has functional groups (phosphate, amino, carboxylate and hydroxyl) attached to it which act as major contributors for degrading dyes present in textile wastewater [49]. The literature demonstrates that three mechanisms are used in algal degradation: (1) switching to non-colored intermediates from dyes; (2) using dyes to support their growth; and (3) chromophores’ uptake on algae. Different algal species have been used for degradation of dyes present in textile wastewater, such as *Anabaena flos-aquae* UTCC64, *chlorella vulgaris*, *Nostoclinckia*, *volvox aureus phormidiumautumnale* UTEX1580, *cosmarium*, *chlorella pyrenoidosa Lyngbyalagerlerimi*, *Elkatothrixviridis* and *Oscillatoriarubescens*. Research shows that algal strains can be used for removing Tartrazine, CR, Remazol black B (RBB), MG, Rhodamine B (RB), basic fuchsin (BF), navy blue HE22 (NBHE22), amido black 10B (ABB10) and methyl red (MR) dyes.

Acquiring the maximal degradation efficiencies of microorganisms for the purpose of decolorizing and mineralizing complex dyes and their resistant metabolites is undoubtedly fraught with difficulties. On the other hand, creating genetically modified microbes can help to partially resolve these problems. By manipulating a bacterial genome in this way, genetic engineering can enable the bacterium to adapt and increase its bioremediation efficiency to levels that are typically beyond the reach of regular bacteria [22].

### 2.5. Bioremediation of Dye Using Yeast Strains

In case of degradation of textile dyes, yeasts have certain advantages over filamentous fungus and bacteria. For instance, they can withstand adverse situations and develop quickly [50]. Studies on yeasts’ decolorization and degradation of dyes are scarce. A yeast strain can extract a significant amount of dye from wastewater.

Yeast strains can degrade textile dyes primarily through biosorption and reductive azo bond cleavage. Yeast is commonly used to decolorize textile dyes, including CR, RB5, acid red B (ARB), RB, MG, and RBB [12]. The effect of pH, time and temperature on degradation of RB5 by *S. halophius* SS-1575 was investigated [13]. After 18 hrs, 100% decolorization efficiency of RB5 was achieved at pH 5. The efficiency of degradation reduced in both acidic and alkaline pH conditions.

### 2.6. Bioremediation of Dye Using Enzymes Strains

Enzymes are liquid biocatalysts. They are selected in order to use precipitation to remove obstinate pollutants [51]. In recent years, biocatalysts have become widely popular for a variety of industrial uses. Compared to conventional methods, enzymatic degradation of textile effluents has several advantages, such as substrate specificity and selectivity, low costs, high efficiency, convenience of use, and green chemistry [52,53]. Biocatalyst deactivation from denaturation is a major challenge in enzymatic [7]. Important operational factors such oxygen transport, pH, temperature, dye structure, redox mediators, dye concentration and enzymes affect the enzymatic decolorization of dyes [54]. Enzyme immobilization is vital for bioremediation of textile dyes in biological reactors. Cross-linking is a popular immobilization approach since it works well with most enzymes [55]. Further immobilization strategies have evolved to better suit various applications, and progress is ongoing. Recent research has examined the efficacy of different enzyme sources for dye degradation. Table 1 enlists various biological strains used for treatment of dyes and the efficiency achieved by each one of them.

Nowadays, cutting-edge technics like bioremediation, which includes bioreactors, nanotechnology, microbial fuel cells, genetic engineering, and others, are being applied in the removal of dye environmental dye pollution shows in Figure 6. A critical description of several methods for eliminating different kinds of dye from wastewater is given below.

## 3. Bioreactor Advancement for Bioremediation of Dyes

Design and development of appropriate bioreactors is crucial for large-scale bioremediation. A bioreactor is a core component of the biological processes used for the removal of various contaminants. It is preferable to carry out an in-depth investigation on the biotic system (such as genetic manipulation, metabolism and cell growth) to establish appropriate experimental parameters (homogenization, operational stability, installation cost, oxygen transfer, shear and scale-up) for the purpose of designing an appropriate bioreactor for a given treatment process [74]. Bioremediation is often carried out in closed bioreactors with carefully controlled settings for optimal results. Several closed bioreactor designs have been developed for improved growth and control over operating parameters. Bioreactors are classified based on their application requirements as semi-batch, batch or continuous operation [75]. Various bioreactor types, including semi-fluidized-bed, moving-bed, fluidized-bed, stirred tank and packed-bed bioreactors, have been used for dye bioremediation based on their intended aerobic and anaerobic applications [76].

Dyes present in textile wastewater can be biodegraded using aerobic or anaerobic, or a combination of the two techniques. To improve process efficiency, biodegradation process or biosorption process should take place in a dedicated reactor, regardless of the technique used. There are various types of bioreactors, such as air-pulsed bioreactors, membrane-based reactors, microbial fuel cell, membrane-based reactors, semi-continuous bioreactors and hybrids. Bioreactors have been extensively examined for their ability to remediate different dyes. Smart distributed sensors and actuators have improved the precision of bioreactor processes, allowing for more efficient control of bioremediation.

### 3.1. Conventional Bioreactors

Conventional bioreactors come in different varieties, each with special advantages and limitations. The types of conventional bioreactors are summarized below.

#### 3.1.1. Membrane Bioreactor

High effluent quality, adaptability, decreased sludge generation, prevention of enzyme washout, and a small footprint are some of the advantages of membrane bioreactors. The membrane’s filtering function efficiently divides liquids and particles, and, hence, the treatment efficiency enhances. Membrane fouling shortens the life of membrane bioreactors. Membrane bioreactor performance has improved drastically in recent times. A membrane bioreactor with the microbial community ‘IHK22’ was used to decolorize effluent from textiles with a decolorization performance of 91–100%. The dye mass loading rates varied from 1.25 to 2.5 mg/g. Berkessa et al. [77] reported an average permeate flow rate of 250 mL/h. In order to eliminate RB5, an aerobic membrane bioreactor (aerobic MBR) was connected with an anaerobic sequencing batch reactor (SBR). Over 99% of the dye was destroyed in 36 h by *Lactococcuslactis* [78]. *Methanomethylovorans* and *Moranbacteria* combined in a membrane bioreactor yielded a maximum rate of 883 mg/(L-day) and approximately 100% MO decolorization efficiency [79].

#### 3.1.2. Stirred Tank Bioreactors(STRs)

STRs are widely utilized for their benefits in temperature control, fluid mixing, oxygen transfer, compliance with good manufacturing practices, low operating costs, ease of scale-up, and alternate impellers [74]. Stirred bioreactors can be used for both batch and continuous processes, and they are reusable. They are available in different sizes. Glass bioreactors are widely used in scientific investigations. Because of their durability, capacity to avoid dead zones, enhanced mixing, and ease of cleaning, cylindrical hemispheres with dished heads are frequently utilized [80]. Stirred tank bioreactors are the favored option due to their constant distribution, improved temperature and pH management, less clogging, improved substrate contact, and effective mixing [81]. An indigo-dyed textile effluent was treated in a continuous stirred tank bioreactor. Within 96 h of operation, the reactor successfully removed 75–80% of COD and color, with the wastewater feed rate ranging from 0.92 to 3.7 g/(L-day) [82]. The usage of STR bioreactors is restricted by a number of issues, such as high shear, high energy consumption, challenges with sealing, and size restrictions because of motor size, length of shaft and weight. To address these issues, novel impeller designs including centrifugal impellers have been developed.

#### 3.1.3. Wave Bioreactors(WBRs)

WBRs use mechanical forward and backward rocking of a bag containing a culture to create a wave motion, effectively mixing and transferring substrate mass [83]. WBRs feature a pre-sterilized, disposable plastic chamber to prevent cross-contamination and improve functioning. Two alternative designs for WBRs have been developed: first, a wave and undertow bioreactor, and second, a slug bubble bioreactor [84]. Both bioreactors can be used in small- to medium-scale cell cultures for many applications.

#### 3.1.4. Airlift Bioreactor

Airlift bioreactors offer a promising alternative for wastewater treatment. Airlift bioreactors offer advantages such as simplicity, low installation cost, no requirement of agitator shaft and low power usage. However, their lack of a mixing mechanism limits mass transfer [12]. *Laccase* enzyme was used in an airlift bioreactor to completely decompose an aromatic dye (IC) under optimal operating conditions. Teerapatsakul et al. [54] used a 5 L airlift bioreactor to decolorize indigo carmine, an aromatic dye, at a concentration of 25 mg/L indigo carmine and a 4 L/min airflow rate. A single enzyme (*laccase*) was observed to be sufficient to achieve 100% dye degradation. *Bjerkanderaadusta* OBR105, a white-rot fungus, was employed in an air-lift bioreactor to decolorize dyes. After three days, most dyes tested showed clearance rates of 91–99% [85].

#### 3.1.5. Fixed-Bed Bioreactor

Textile effluent decolorization appears to be a promising use for fixed/packed-bed bioreactors. Real textile effluent was successfully decolorized by a bacterial–yeast consortium using a triple-layered fixed-bed reactor. At a flow rate of 100 mL/h, the reactor achieved more than 80% decolorization over a period of 7 days and COD was also decreased by about 78% [86].

#### 3.1.6. Fluidized Bed Bioreactor

These reactors offer numerous advantages, including consistent particle mixing, temperature gradient control, and continuous operation. However, several disadvantages, such as a larger reactor volume, requirement of pumping owing to pressure drop, entrapment of particles, and internal part degradation, limit their use. A study on acid yellow 36 decolorization in a recirculating semi-pilot system examined the efficiency of fluidized bed Fenton oxidation [87]. To eliminate CR, a multistage fluidized bed bioreactor (M-FBBR) was used, with PUF-immobilized *Bacillus subtilis* [88].

#### 3.1.7. Modern Bioreactors

Removing resistant colors from wastewater requires the combination of more than one treatment approach. Dye degradation products and excessive salinity can significantly reduce cell activity and metabolism, hindering biological treatments [89]. Using combination or hybrid treatment technologies improves efficiency and meets regulatory criteria for wastewater treatment [6]. Using several treatment methods on a continual basis can lead to complications and longer processing times. Hybrid methods, combining different treatment protocols, may be the most effective solution. Hybrid methods save time and money by utilizing a single vessel for various processes. Compared to hybrid processes, integrated processes require multiple phases and can be costly and time-consuming [6,90]

#### 3.1.8. Combined or Sequential Bioreactors

Textile dye degradation can be achieved by a variety of chemical/biological treatment techniques. Various combinations exist, such as a down-flow hanging sponge along with cationic polymer (Organ polymer), coagulation/flocculation followed by sequencing batch reactor, up-flow biological aerated filtration (UBAF) with zonation, and ozonation combined with biofilm [22]. Using ozonation alone to treat dye effluent can produce carcinogenic by-products, increasing toxicity [91]. To solve this problem, the authors employed a biological treatment using biofilm following ozonation. Heavy organic loads, absence of washout or clogging problems, a long biomass retention time, effective COD reduction, and a compact reactor design are just a few advantages of combining chemical and biological processes. The simultaneous decolorization and biodegradation of dye-containing textile effluent is an advantage of combined biological and biological treatments. By breaking down dye compounds, bacteria can absorb C, O, N, S and other components, reducing the requirement for extra nutrition [6]. Complex dye compounds may take longer to decolorize and degrade despite the higher efficiency of these reactors.

#### 3.1.9. Hybrid Bioreactors

Hybrid treatment techniques outperform single-degradation procedures, as previously stated. Azo dyes must be broken down anaerobically by an organic co-substrate that has the potential to supply an electron needed to break the dye’s azo bonds [92]. Hybridizing electrolysis with anaerobic degradation, known as microbial electrolysis, can improve microbial activity through electrical stimulation [93]. Recent research suggests that an anaerobic reactor with zero-valent iron could be a useful tool for treating azo dye wastewater. As a reducing agent, the reactor employed zero-valent iron (ZVI) in an up-flow anaerobic sludge blanket. Decolorization efficiency of 91.7% was attained, in contrast with 43.8% and 28% for the zero-valent iron and up-flow anaerobic sludge blanket treatments, respectively [94]. In another study, Liu et al. [70] used scrap iron and graphite electrodes in an anaerobic reactor to process dye. The new approach improved dye degradation and COD reduction, but also resulted in scrap iron bed blockage. The use of Fe-graphite plate electrodes resulted in 83.4% discoloration efficiency and 84.7% COD reduction for azo dye-containing wastewater at a concentration of 1200 mg/L [95].

Vikrant et al. [22] used integrated an MBR, which consisted of thermophilic, anoxic/aerobic, biofilm and bioaugmented MBRs in order to improve removal efficiency. A hollow fiber module with GAC coating EMR effectively degraded acid orange II, with 85% to nearly complete dye degradation. Wang used a combined bioreactor which was acidogenic in nature to successfully degrade the azo dye acid red G.

## 4. Microbial Fuel Cell for Removal of Dyes

Microbial fuel cells (MFCs) have been widely used to cleanse wastewater containing azo dyes and generate power using microorganisms as biocatalysts. Conventional dye treatment methods, as previously discussed, have a number of drawbacks, including the need for enormous amounts of energy and chemicals, as well as challenges with processing and removing produced sludge and secondary waste streams. Although they are ineffective for full dye breakdown and result in toxic by-products, biological therapy approaches have been promoted as a practical and affordable substitute [96,97]. To speed up anaerobic bioremediation of dyes, an electron (e^−^) donor (organic co-substrate) is necessary [98]. Determining the precise amount of necessary co-substrate by stoichiometry can be quite intricate, resulting in an overabundance of e-donor addition that raises operating expenses and generates pollutants such as methane [22]. Biological electrochemical systems use electroactive bacteria as biocatalysts to generate electricity. Organic co-substrate is oxidized at the anode by microorganisms in MFCs. Microorganisms transmit electrons generated during oxidation to the anode through complicated extracellular pathways. The electrons travel to the cathode, generating an electric current. Coupled dye removal involves transport of protons to the cathode through an ion exchange membrane, which is the place of combination of electrons and azo dyes to reduce the azo bond [22].

### 4.1. Types of MFC

#### 4.1.1. Single-Chamber MFC

A single-chamber MFC is a straightforward and economical design approach [99]. It is composed of a basic anode chamber and an air-cathode based on microfiltration membranes [100]. The inner and outer sides of the cathode are oriented toward water and air, respectively. A PEM is not necessary for this arrangement [99]. Solanki et al. [101] suggest placing a microfiltration membrane directly on the cathodic inner surface. According to Khalili et al. [100] and Solanki et al. [101], the cathode is covered with a thick piece of plexiglas or a comparable material. To transfer oxygen to the cathode, the cover has [100]. A combination of anaerobic and aerobic sludge makes up the inoculum. A single-chamber MFC experiences azo bond cleavage due to protons and electrons generated from oxidation of substrate at the anode by action of microbes, much like dual-chamber MFCs.

#### 4.1.2. Dual-Chamber MFCs

An MFC uses a proton exchange membrane for separately keeping the anaerobic anode and aerobic cathode inside the chamber [102]. While the anode inoculum consists of a merger of substrate and anaerobic sludge (such as glucose), the cathode inoculum is aerobic sludge [103]. As per Solanki et al. [101], the pH of each chamber is maintained at 7. Liu et al. [102] and Zhang et al. [102] describe how microbes in the anodic chamber oxidize dye molecules to release protons and electrons. While protons move through the PEM, electrons move via the external circuit. According to Cui et al. [2] and Zhang et al. [103], the electron acceptor combines with the electron and the proton at the cathode.

### 4.2. MFC Microorganisms

Most microorganisms lack electrochemical activity [40,99]. In MFCs, mediators such as methyl viologen, methyl blue, neutral red and thionine can help transfer e^−^ to the anode generated by microorganisms [99,100]. However, the usage of mediator-based MFCs is limited because of their costly nature and toxicity [40]. Modern MFCs rely on electrochemically active microorganisms that thrive in anodic environments. Electrochemically active microbes in MFCs include *Bacillus*, *Geobacter and Enterobacter* [104,105] and *Shewanella bacteria* [106]. Mixed bacterial consortiums are preferred over pure cultures due to their superior stability, stress resistance, and nutritional adaptation [103]. Mixed-bacteria-based MFCs exhibit higher current density with respect to unadulterated culture-based MFCs [101]. According to Khalili et al. [100], the variation in electron transport mechanisms between pure strains and mixed cultures can lead to a reversed trend. The same circumstances must be met in order to compare the physiological and mechanistic characteristics of the two communities of microbes present in MFCs [40,100].

### 4.3. MFC-Based Bioremediation Mechanism

Dye degradation in bioreactors involves enzymes, sulfide reduction, and low-molecular-weight redox mediators [100,107]. Dyes experience anaerobic oxidation at the anode, as previously mentioned. According to Yang et al. [108], every mechanism used in the decolorization processes is comparable to that of conventional bioremediation, with the exception of the circuitry, which is external, and membranes incorporated in MFCs provide additional means of electrons (e^−^) and proton transfer up to dyes. Yet, in nearly every instance, products mostly consist of amines and sulfanilic acids [101,107].

In the process of bioremediation of acid orange 7 (AO7) dyes, the azo bonds present in the molecules of dyes are separated at the cathode by utilizing protons and electrons secreted at the anode. As a result, dangerous intermediaries are created and then broken down by abiotic mechanisms [107]. Similarly, considering the case of Congo red, the anodic chamber is where the dye decolorizes biologically, while the cathodic compartment is where the intermediates break down abiotically, resulting in total degradation [109,110]. In addition to the power generated during the co-metabolism of glucose and dye molecules, comparable findings have been reported [8]. Aerobic treatments were used to remove the generated AAs and other pathogenic intermediates.

### 4.4. Future Scope and Challenges Associated with MFCs

Several studies with pure and disparate cultures of microbes have been investigated. Majority of these studies indicate that e^−^ are provided by the co-substrates for the dyes’ bioremediation and the concurrent production of energy [40,111]. The common rivalry between e^−^ can be greatly decreased by employing contaminated water as the co-substrate and introducing species which are electrochemically active (e.g., *Geobacter* and *Enterobacter*) to the microbial population. The single-chambered architecture of the MFC is favored because it can simultaneously create power and remediate dyes. This is due to the fact that single-chambered MFCs consume less energy and have a lower internal resistance as compared to dual-compartment MFCs [112,113].

The appropriate mechanisms of the anaerobic process and the corresponding MFCs’ columbic efficiency in a single-chamber design are severely affected by the transport of oxygen from cathodic to anodic compartments [101]. These disadvantages can be reduced by choosing a more air-resistant polymer to cover the microfiltration membrane [113]. Currently, the low power production and very expensive operation of MFCs limit their practical applicability [40]. The electrode’s effective design is anticipated to lower the MFCs’ cost. The potential application of metal- and carbon-based versions as potential electrodes has been investigated recently [114,115]. With the advent of sophisticated new configurations of electrode, like a bio-cathode that uses microbes to help with assisted electron transfer, the requirement for noble metals such as platinum and renewable e-mediators is reduced, improving MFC sustainability [2,115].

Furthermore, because they can provide a sizable area for the development of biofilms with enhanced activity of MFC, granular activated carbon bio-cathodes (GACs) have also been suggested as very promising substitutes of conventional electrodes [116,117]. GAC-based MFCs are incredibly adaptable and do not require any particular pre-treatment, pH adjustment, or external electron donors. Cost-effective electrode designs and the optimization of various parameters can aid in the large-scale development of MFCs [118,119].

## 5. Genetically Engineered Microorganisms (GEMs)

Genetically engineered microorganisms (GEMs) are microbes whose genetic material has been modified to enhance their capacity to degrade, transform, or remove pollutants, including synthetic dyes from industrial effluents (especially textile wastewater) [120]. Genetic engineering uses genetic techniques to selectively modify microbes (such as bacteria, fungus, and yeast) [121]. In their investigation, Bu et al. [122] carried out dye decolorization tests using wild-type laccase, D500G, and mutant laccase Lacep69. They found that D500G had a 78% decolorization rate for acid violet, which was superior to that of wild-type laccase. Dixit and Garg [123] successfully decolorized 95% of azo dyes in less than 24 h by transferring the azoreductase enzyme-coding gene azoK from Klebsiella pneumonia to Escherichia coli. Chang et al. [124] stated that by modifying the E. coli genes that code for the azoreductase found in Pseudomonas luteola, they were able to increase the biodecolorization of azo dyes by 10%.

## 6. Nanoparticle-Based Bioremediation

Nanoparticle-based bioremediation is an emerging field that combines nanotechnology with microbial or enzymatic systems to enhance the degradation, detoxification, or removal of pollutants from contaminated environments (soil, water, and air). This technique uses nanoparticles (between 1 and 100 nm) either alone or in synergy with microorganisms or enzymes to improve the efficiency, speed, and specificity of bioremediation [125,126]. Due to their high surface-area-to-volume ratio, reactivity, and surface functionalization capabilities, they are highly effective in interacting with pollutants. Balrabe et al. [126] found that eosin yellowish was effectively removed (95%) at 75 ppm using Fe_2_O_3_ nanoparticles at pH 6 after 15 min of exposure. Another work by Darwesh et al. [127] found that CuO-NPs produced by the F. oxysporum OSF18 strain and bound in alginate beads demonstrated 90% dye remediation effectiveness. The remediation of organic dyes using functionalized nanoparticles was reported by Rani and Shanker [128]. To further understand the role of nanomaterials in large-scale in situ bioremediation, more research is required.

## 7. Comparisons of Different Bioremediation Techniques

In bioremediation of dye-contaminated wastewater, the focus shifts specifically to methods capable of degrading complex dye molecules, especially synthetic dyes (e.g., azo, anthraquinone, and triphenylmethane dyes), which are toxic, recalcitrant, and non-biodegradable under conventional treatments [129,130]. The comparison of different bioremediation techniques specifically for dye-contaminated wastewater, analyzed in terms of advantages, scalability, economic feasibility, is shown in Table 2.

For large-scale industrial dye wastewater, bacterial bioremediation in bioreactors is the most scalable and efficient. For eco-sensitive or low-budget scenarios, constructed wetlands or algal systems are ideal. Enzyme-based and fungal treatments are promising but currently best suited for niche or small-scale applications due to the cost and operational complexity.

## 8. Conclusions

Various physicochemical treatment techniques have been implemented thus far to lower the total amounts of dye pollution in water bodies. However, the high operational and energy costs, sludge generation, harmful by-products, huge chemical requirements and energy penalties limit the effectiveness of all these conventional techniques. This review outlined several biotechnological methods for successfully removing dyes from water. Due to a number of recognized benefits, bioremediation techniques are suitable options for dye removal.

## Figures and Tables

**Figure 1 bioengineering-12-01043-f001:**
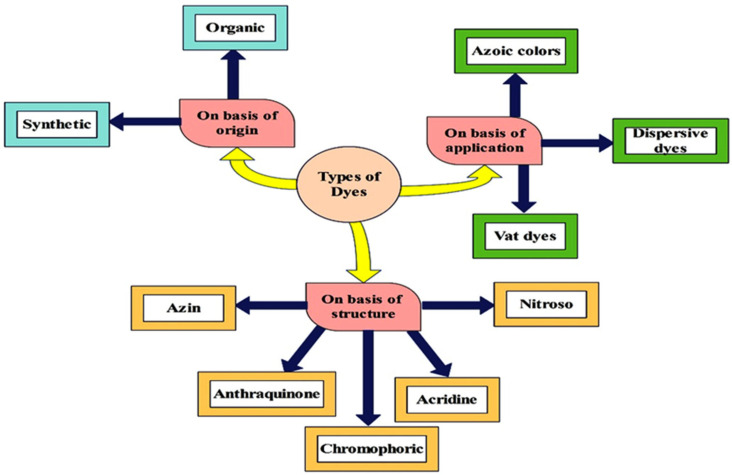
Classification of dyes on the bases of origin, application and structure.

**Figure 2 bioengineering-12-01043-f002:**
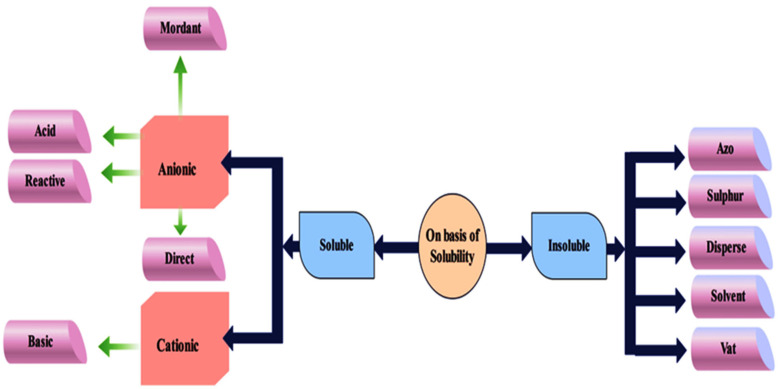
Classification of dyes on the basis of solubility.

**Figure 3 bioengineering-12-01043-f003:**
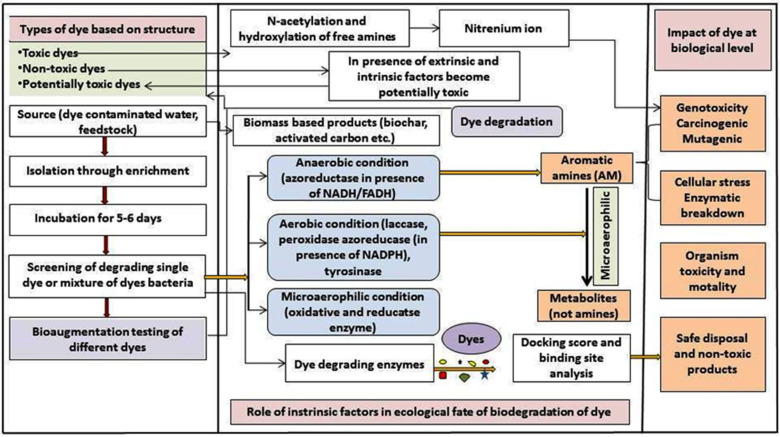
An integrated approach to dye interaction with various degradation techniques.

**Figure 4 bioengineering-12-01043-f004:**
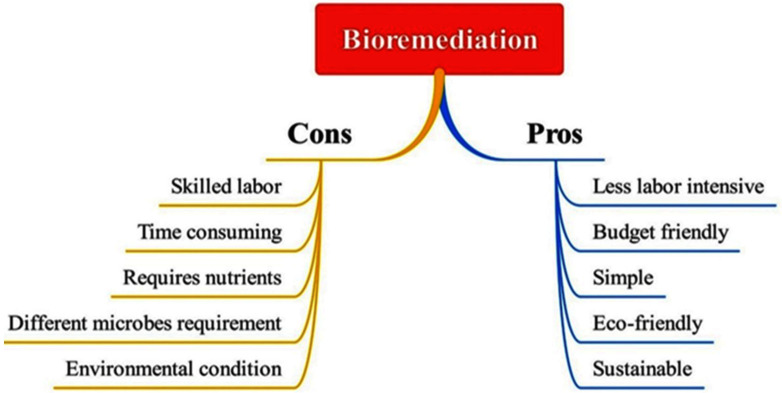
Pros and cons of bioremediation techniques.

**Figure 5 bioengineering-12-01043-f005:**
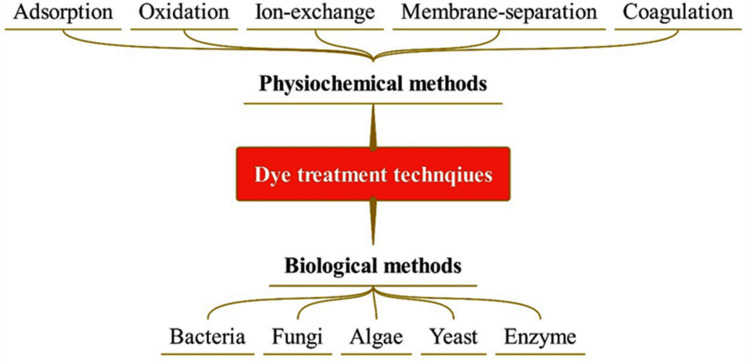
Various kinds of bioremediation techniques.

**Figure 6 bioengineering-12-01043-f006:**
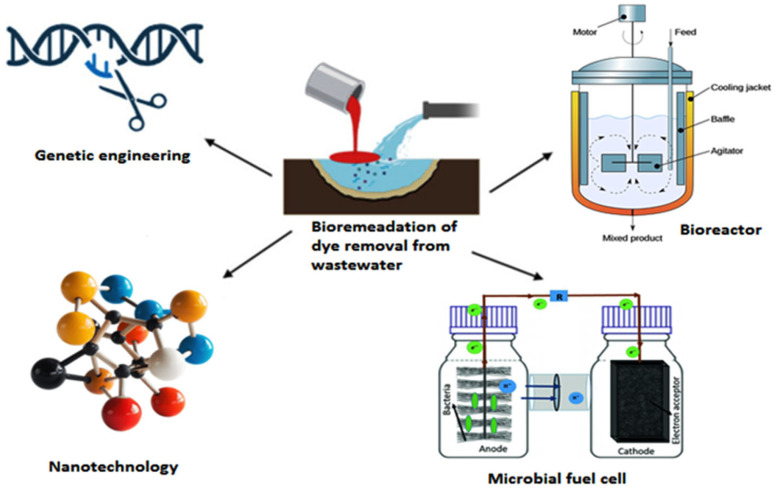
Current approaches to the removal of dyes from aquatic environments.

**Table 1 bioengineering-12-01043-t001:** Performance of various biological species for dye removal.

Type of Species	Name of Species	Dye	Optimum Operating Conditions (Static)	Peak Removal Efficiency	Ref.
Bacteria	*Pseudomonas entomophilaBS1*	Reactive black 5	5–9 pH, 37 °C, 120 h, 500 mg/L	93%	[19]
	Consortium of *pseudomonas species* SUK1 and *pseudomonas rettgeri* strain HSL1	Reactive orange 16	30 ± 02 °C 18–24 h, 100 mg/L	100%	[37]
		Disperse red 78	30 ± 02 °C 36–42 h, 100 mg/L	100%	[37]
		Reactive black 5	30 ± 02 °C 48 h, 100 mg/L	58%	[37]
		Direct red 81	30 ± 02 °C 48 h, 100 mg/L	92%	[37]
	DDMY1	Reactive black 5	5.0 pH, 100 mg/L 30–40 °C	87.45 ± 1.09%	[38]
		Reactive black 5	9.0 pH, 100 mg/L 30–40 °C	88.89 ± 2.56%.	[38]
Bacteria	*Aeromonashydrophila*	Reactive black 5	24 h, 100 mg/L	76%	[56]
	*Bacillus cereus* (MTCC 9777)	Acid orange 7	pH 8.0, 96 h, 100 mg/L	52.5%	[20]
	*Kocuriarosea* MTCC 1532	Methyl orange	6.8 pH, 30 °C, 50 mg/L	100%	[57]
	*Pseudomona* ssp. *SUK1*	Reactive red 2	6.2–7.5 pH, 30 °C, 24 h, 5 mg/L	91%	[58]
	*Pseudomonas* *aeruginosa* NBAR12	Reactive blue 172	7.0 pH, 40 °C 42 h, 500 mg/L	83%	[59]
	*Lysinibacillus* sp. *RGS.*	C.I. Remazol red	7.8 pH, 30 °C, 48 h, 50 mg/L	87%	[60]
	*Bacillus* sp.VUS	Red HE7B	18 h, 50 mg/L	100%	[61]
Bacteria- yeast consortium	*Brevibacilluslaterosporus* MTCC 2298—*Galactomycesgeotrichum* MTCC1360	Scarlet RR	9.0 pH, 40 °C, 18 h, 50 mg/L	98%	[62]
Fungus	*Penicilliumoxalicum*(SAR-3)	Direct red 75, direct blue 15 and acid red 183	7.0 pH, 30 °C, 120 h, 100 mg/L	96.6 ± 3.25%	[41]
Fungus	Laccase from *Peroneutypascoparia*	Acid red 97	6.0 pH, 40 °C, 84 h, 100 mg/L	75%	[63]
	Bacilluscereus (MTCC 9777) RMLAU1	Acid orange 7	8.0 pH, 33 °C, 96 h, 100 mg/L	68.5%	[20]
	mycelium pellets of *Penicilliumoxalicum*	Reactive blue 19	2.0 pH, 20 °C, 100 mg/L	91%	[64]
	*Rhizopusoryzae* MTCC 262	Rhodamine B	7.0 pH,40 °C, 05 h, 100 mg/L	90%	[65]
	*Neurosporacrassa*	Acid red 57	1.0 pH, 20 °C, 40 min, 100 mg/L	98.7%	[45]
	*Curvularia clavate* NZ2	Congo red (CR)	5.0 pH, 05 h, 100 mg/L	96.1 ± 1%	[44]
	*Curvularia clavate* NZ2	RB5	5.0 pH, 05 h, 100 mg/L	90.3 ± 1.86%	[44]
	*Curvularia clavate* NZ2	Acid orange 7 (AO7)	5.0 pH, 05 h, 100 mg/L	46.3 ± 1.86%	[44]
	*Lasiodiplodia so.*	Malachite green	7.0 pH,30 °C, 24 h, 50 mg/L	96.9%	[46]
Algae	*Hydrocoleumoligotrichum*	Basic fuchsin	7 days, 5 mg/L	92.44%	[66]
Algae	*Oscillatorialimnetica*	Basic fuchsin	7 days, 5 mg/L	90.23%	[66]
	*Hydrocoleumoligotrichum*	Methyl red	7 days, 20 mg/L	53.23%	[66]
	*Oscillatorialimnetica*	Methyl red	7 days, 20 mg/L	50.18%	[66]
	*Chlorella vulgaris*	Remazol black B	2.0 pH, 35 °C, 800 mg/L	53.2%	[67]
	*Sargassumhorneri*	Methylene blue	5.0–5.5 pH, 25 °C, 02 h, 200 mg/L	92.5%	[68]
	*Sargassumhorneri*	Methylene blue	5.0–5.5 pH, 25 °C, 02 h, 200 mg/L	89.7%	[68]
	*Ulvaaustralis*	Toluidine blue	5.0–5.5 pH, 25 °C, 02 h, 200 mg/L	95.3%	[68]
	*Ulvaaustralis*	Toluidine blue	5.0–5.5 pH, 25 °C, 02 h, 200 mg/L	96.4%	[68]
	*Ulvafasciata*	Methylene blue	04 h, 100 mg/L	88.9%	[69]
	*Ulvafasciata*	Congo red	04 h, 50 mg/L	79.6%	[69]
	*Sargassumdentifolium*	Methylene blue	30 min, 100 mg/L	82.1%	[69]
Algae	*Sargassumdentifolium*	Congo red	04 h, 100 mg/L	85%	[69]
Yeast	*Sterigmatomyces* *halophilus* SSA1575	Reactive black 5	5.0 pH, 30 °C, 18 h, 50 mg/L	100%	[13]
	*Candida rugopelliculosa* HXL-2	Reactive blue 13	5.0 pH, 28 °C, 28 h, 50 mg/L	90%	[70]
Enzyme	*Irpexlacteus* F17	Malachite green	3.1 pH, 40 °C, 24 h, 200 mg/L	96%	[71]
	Nanocellulose immobilized laccase enzyme (PersiLac1)	Malachite green	5.0 pH, 50 °C, 24 h, 150 mg/L	98%	[72]
	Nanocellulose immobilized laccase enzyme (PersiLac1)	Congo red	5.0 pH, 50 °C, 24 h, 150 mg/L	60%	[72]
	*Citrus limon* peroxidase	1847 Colafx blue P3R	4.0 pH, 35 °C, 1 h, 200 mg/L	83%	[73]
	*Citrus limon* peroxidase	621 Colafx blue	4.0 pH, 35 °C, 1 h, 200 mg/L	99%	[73]

**Table 2 bioengineering-12-01043-t002:** Comparison of different bioremediation techniques [53].

Technique	Advantages	Scalability	Economic Feasibility
**ASP**	Well-studied, effective	High	Moderate
**Algal**	Eco-friendly, multi-nutrient removal	Moderate	Moderate
**Fungal**	Effective for complex dyes	Moderate	Moderate
**Bacterial**	Fast, adaptable	High	High
**Bioreactors**	Controlled, efficient, compact	High	Moderate
**Wetlands/phyto**	Low-maintenance, sustainable	Moderate	High
**Enzyme-based**	High specificity, no biomass	Low–Moderate	Low
**Algal**	Eco-friendly, multi-nutrient removal	Moderate	Moderate

## Data Availability

The authors confirm that the data supporting the findings of this study are available within this article.

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
