# Peer review of "Environmental Impacts and Strategies for Bioremediation of Dye-Containing Wastewater"

_bioengineering, 2025, doi:10.3390/bioengineering12101043_

Round 1
Reviewer 1 Report
Comments and Suggestions for Authors
This review article “Environmental impacts and strategies for bioremediation of dye-containing wastewater” is well-structured and informative for the readers. A few points to be addressed before acceptance.
Questions to authors:
- Request the authors to mention the time frame (e.g., last 10 years) of the literature surveyed.
- Several comprehensive reviews on dye wastewater treatment and bioremediation are already available in the literature (e.g., Aravin Prince Periyasamy, 2025 https://doi.org/10.1016/j.clwat.2025.100092 , Rania Al-Tohamy et al, 2022 https://doi.org/10.1016/j.ecoenv.2021.113160 , Yuqi Liu et al., 2024 https://doi.org/10.3390/su162410867). To strengthen the manuscript, please include these reviews & clarify how this review provides new insights compared to these earlier works.
- This manuscript discusses aerobic and anaerobic degradation of dyes in detail; however, the description is largely textual. To improve clarity, request the authors to provide a schematic or flow diagram that illustrates the key steps and intermediates in aerobic versus anaerobic dye degradation pathways to help readers in comparing the mechanisms.

Author Response
Point-by-point response to Comments and Suggestions for Authors
Summary: Dear Sir, first of all, thanks for reviewing the manuscript. Your insightful comments helped us a lot to enhance the quality of the manuscript to a large extent. We have incorporated your suggestions and revised the manuscript. All the changes are highlighted in yellow. We hope that the revised manuscript is in line with your expectations and as per the journal standards.
Comments 1: Request the authors to mention the time frame (e.g., last 10 years) of the literature surveyed.
Response 1: Dear reviewer, this review paper mainly focuses on a time frame of last five years. However, we have also tried to include relevant literature from the time duration of 2015-2019. In order to enrich literature review these recent references have been added in the revised manuscript. The changes are highlighted in yellow.
Comments 2: Several comprehensive reviews on dye wastewater treatment and bioremediation are already available in the literature (e.g., Aravin Prince Periyasamy, 2025 https://doi.org/10.1016/j.clwat.2025.100092, Rania Al-Tohamy et al, 2022 https://doi.org/10.1016/j.ecoenv.2021.113160,Yuqi Liu et al., 2024 https://doi.org/10.3390/su162410867). To strengthen the manuscript, please include these reviews & clarify how this review provides new insights compared to these earlier works.
Response 2: Dear reviewer, the manuscript has been revised as per your kind suggestion. We have added all the references which are mentioned in this comment and also added some references with relevant earlier works. Changes are highlighted in yellow.
- Periyasamy A.P. A review of bioremediation of textile dye containing wastewater.Cleaner Water. 2025, 4, 100092.https://doi.org/10.1016/j.clwat.2025.100092.
- Al-Tohamy, R.; Ali, S.S.; Li, F.; Okasha, K.M.; Mahmoud, Y.A.G.; Elsamahy, T.; Jiao, H.; Fu, Y.; Sun, J. A critical review on the treatment of dye-containing wastewater: Ecotoxicological and health concerns of textile dyes and possible remediation approaches for environmental safety.Ecotoxicology and Environ.Safety.2022, 231, 113160. https://doi.org/10.1016/j.ecoenv.2021.113160.
- Liu, Y.; Chen, J.;Duan, D.; Zhang, Z.; Liu, C.; Cai, W.; Zhao, Z. Environmental Impacts and Biological Technologies Toward Sustainable Treatment of Textile Dyeing Wastewater: A Rev., Sustaina.2024, 16(24),https://doi.org/10.3390/su162410867.
- Yang, Y.; Zhang, W.; Zhang, Z.; Yang, T.; Xu, Z.; Zhang, C.; Guo, B.; Lu, W. Efficient Bioremediation of Petroleum-Contaminated Soil by Immobilized Bacterial Agent of Gordonia alkanivoransBioengineering. 2023, 10(5), 561.
https://doi.org/10.3390/bioengineering10050561.
5. Sravan, J.S.; Matsakas, L,;Sarkar, O.P.Advances in Biological Wastewater Treatment Processes: Focus on Low-Carbon Energy and Resource Recovery in Biorefinery Context. Bioengineering. 2024, 11(3), 281. https://doi.org/10.3390/bioengineering11030281.
6. Ruben A.; Martins, Salgado, E.M.;Gonçalves, A.L.; EstevesandJose, A.F.; Pires, C.M. Microalgae-Based Remediation of Real Textile Wastewater: Assessing Pollutant Removal and Biomass Valorisation. Bioengineering. 2024, 11(1), 44.
https://doi.org/10.3390/bioengineering11010044.
- Tripathi, M.; Singh, S.; Pathak, S.; Kasaudhan, J.; Mishra, A.; Bala, S.; Garg, D.; Singh, R.; Singh, P.; Singh, P.K.; Shukla, A.K.; Pathak, N. Recent Strategies for the Remediation of Textile Dyes from Wastewater: A Systematic Review. 2023, 11, 940.https://doi.org/10.3390/toxics11110940.
- Zahuri, A.A.; Abdul Patah, M.F.; Kamarulzaman, Y.; Hashim, N.H.; Thirumoorthi, T.; Wan Mohtar, W.H.M.; MohdHanafiah, Z.; Amir, Z.; Wan-Mohtar, W.A.A.Q.I. Decolourisation of Real Industrial and Synthetic Textile Dye Wastewater Using Activated Dolomite. 2023, 15, 1172.https://doi.org/doi: 10.3390/w15061172.
- Harper, R.; Moody, S.C. Filamentous Fungi Are Potential Bioremediation Agents of Semi-Synthetic Textile Waste. Fungi.2023, 9, 661.
https://doi.org/10.3390/jof9060661.
- Periyasamy, A.P. Recent Advances in the Remediation of Textile-Dye-Containing Wastewater: Prioritizing Human Health and Sustainable Wastewater Treatment. Sustainability. 2024, 16, 495. https://doi.org/10.3390/su16020495.
- Xie, R.; Danso, B.; Sun, J.; Al-Zahrani, M.; Dar, M.A.; Al-Tohamy, R.; Ali, S.S. Biorefinery and Bioremediation Strategies for Efficient Management of Recalcitrant Pollutants Using Termites as an Obscure yet Promising Source of Bacterial Gut Symbionts: A Review 2024, 15, 908.
https://doi.org/doi: 10.3390/insects15110908.
- Singh, G.; Chaudhary, S.; Giri, B. S.; Mishra, V. K. Assessment of geochemistry and irrigation suitability of the River Ganga, Varanasi, India: PCA reduction for water quality index and health risk evaluation. Environ Sci Pollut Res 2025, 32, 4199–4218. https://doi.org/10.1007/s11356-025-35912-8.
- Tiwari, S.K.; Giri, B.S., Thivaharan, V. et al.Sequestration of simulated carbon dioxide (CO2) using churning cementations waste and fly-ash in a thermo-stable batch reactor (TSBR). Environ Sci Pollut Res 27, 27470–27479 (2020). https://doi.org/10.1007/s11356-019-07342-w.
Comments 3: This manuscript discusses aerobic and anaerobic degradation of dyes in detail; however, the description is largely textual. To improve clarity, request the authors to provide a schematic or flow diagram that illustrates the key steps and intermediates in aerobic versus anaerobic dye degradation pathways to help readers in comparing the mechanisms.
Response 3: Dear reviewer, thanks for your insightful suggestion. We have incorporated a schematic or a flow diagram (Figure 3) that illustrates the key steps and intermediates in aerobic versus anaerobic dye degradation pathways to help readers in comparing the mechanisms. Changes are highlighted in yellow.
Hence, we request you to please reconsider this revised manuscript for the consideration of the possible publication in your prestigious journal.

Reviewer 2 Report
Comments and Suggestions for Authors
This manuscript provides a comprehensive review on the environmental impacts of dye-containing wastewater and the strategies for bioremediation using microorganisms, enzymes, algae, fungi, yeast, and advanced bioreactors and microbial fuel cell (MFC) systems. The topic is highly relevant given the increasing concerns about water pollution and the urgent need for sustainable treatment technologies. However, the manuscript is written in a rather superficial manner. Therefore, only with substantial improvements—streamlining, deeper critical analysis, and stronger emphasis on recent advances—could it be considered for publication.
1. Some sections (1.3, 2.1) contain repeated descriptions, particularly regarding the limitations of conventional methods. These should be condensed to improve readability. A more concise narrative would strengthen the review.
2. While many bioremediation approaches are described, the manuscript often stops at reporting outcomes. A more critical synthesis is needed, comparing the relative advantages, scalability, and economic feasibility of different techniques. For example, which microbial systems show the highest potential for industrial application?
3. Although microbial fuel cells and bioreactors are discussed, other emerging areas—such as synthetic biology, genetically engineered microorganisms, photocatalysis–biological hybrids, and nanomaterial-assisted systems—are only briefly mentioned. Expanding these sections would better highlight the novelty of the review.
4. Figures 1–4 mainly illustrate classifications and general concepts but do not provide substantial analytical value. Comparative charts (e.g., performance of different strains under similar conditions) would be more informative for readers.
5. The overall quality of the figures is not suitable for scientific publication. They do not clearly convey their intended message, nor are they visually engaging. Moreover, Figures 1 and 2 are merged in a way that appears cut off, as are Figures 3 and 4, which makes them even harder to interpret. The figures require extensive revision.
6. Table 1 is very detailed but somewhat overwhelming. A summary table highlighting only the top performing conditions would improve accessibility.
7. The manuscript would benefit greatly from language polishing to improve clarity and flow. Many sentences are overly long or repetitive, which affects readability. A thorough language edit by a fluent English speaker is strongly recommended.
Author Response
Comments 1: Some sections (1.3, 2.1) contain repeated descriptions, particularly regarding the limitations of conventional methods. These should be condensed to improve readability. A more concise narrative would strengthen the review.
Response 1: Dear reviewer, the manuscript (section 1.3 and 2.1) has been revised as per your kind suggestion. We have corrected and rewrite the sentences particularly regarding the limitations of conventional methods. Changes are highlighted in yellow.
Comments 2: While many bioremediation approaches are described, the manuscript often stops at reporting outcomes. A more critical synthesis is needed, comparing the relative advantages, scalability, and economic feasibility of different techniques. For example, which microbial systems show the highest potential for industrial application?
Response 2: Dear reviewer, as per your kind suggestion, we have incorporated a comparison table with relative advantages, scalability, and economic feasibility of different techniques (Table 2). The changes are highlighted in yellow.
Comments 3: Although microbial fuel cells and bioreactors are discussed, other emerging areas-such as synthetic biology, genetically engineered microorganisms, photocatalysis–biological hybrids, and nanomaterial-assisted systems-are only briefly mentioned. Expanding these sections would better highlight the novelty of the review.
Response 3: Dear reviewer, we have revised the manuscript as per your kind suggestion (section 5 and 6) has been added. The changes are highlighted in yellow.
Comments 4: Figures 1–4 mainly illustrate classifications and general concepts but do not provide substantial analytical value. Comparative charts (e.g., performance of different strains under similar conditions) would be more informative for readers.
Response 4: Dear reviewer, we have tried to maintain our focus on the performance of various treatment strategies/approaches. However, we have incorporated your kind suggestion in the revised manuscript. The changes are highlighted in yellow.
Comments 5: The overall quality of the figures is not suitable for scientific publication. They do not clearly convey their intended message, nor are they visually engaging. Moreover, Figures 1 and 2 are merged in a way that appears cut off, as are Figures 3 and 4, which makes them even harder to interpret. The figures require extensive revision.
Response 5: Dear reviewer, figure 1 & 2 represent the classification of dye on different basis (Figure 1: on basis of origi and Figure 2: on basis of solubity). However, the quality of all the figures has been updated as per your kind suggestion. The changes are highlighted in yellow.
Comments 6: Table 1 is very detailed but somewhat overwhelming. A summary table highlighting only the top performing conditions would improve accessibility.
Response 6: Dear reviewer, Table 1 has been revised as per your kind suggestion. The changes are highlighted in yellow.
Comments 7: The manuscript would benefit greatly from language polishing to improve clarity and flow. Many sentences are overly long or repetitive, which affects readability. A thorough language edit by a fluent English speaker is strongly recommended.
Response 7: As per kind suggestions of reviewer, Language of whole manuscript has now been thoroughly checked once again for the improvement in grammatical errors. The changes are highlighted in yellow.
Hence, we request you to please reconsider this revised manuscript for the consideration of the possible publication in your prestigious journal.

Round 2
Reviewer 2 Report
Comments and Suggestions for Authors
After completing the review, I find the manuscript suitable for publication.